# Immobilization of Enological Pectinase on Magnetic Sensitive Polyamide Microparticles for Wine Clarification

**DOI:** 10.3390/foods13030420

**Published:** 2024-01-28

**Authors:** Sandra Cristina Oliveira, Nadya Vasileva Dencheva, Zlatan Zlatev Denchev

**Affiliations:** IPC—Institute for Polymers and Composites, Campus of Azurém, University of Minho, 4800-058 Guimarães, Portugal; b8223@dep.uminho.pt (S.C.O.); denchev@dep.uminho.pt (Z.Z.D.)

**Keywords:** rosé wine clarification, pectinase, noncovalent enzyme immobilization, microparticulate enzyme supports

## Abstract

The use of free pectinases as clarification biocatalysts constitutes a well-established practice in the large-scale production of various types of wines. However, when in the form of free enzymes, the recovery and reusability of pectinases is difficult if not impossible. To address these limitations, the present study focuses on the noncovalent adsorption immobilization of a commercial pectinolytic preparation onto highly porous polyamide 6 (PA6) microparticles, both with and without magnetic properties, prepared via activated anionic polymerization. The two pectinase complexes resulting after immobilization underwent comparative activity and kinetic studies, contrasting them with the free enzyme preparation. In comparison with the free enzyme, the PA6-immobilized pectinase complexes exhibited more than double the specific activity toward the pectin substrate. They displayed a slightly higher affinity to the substrate while acting as faster catalysts that were more resistant to inhibition. Furthermore, the immobilized complexes were applied in the clarification process of industrial rosé must, whereby they demonstrated accelerated performance as compared with the free enzyme. Moreover, the PA6-immobilized pectinase biocatalysts offered the potential for three consecutive cycles of reuse, achieving complete rosé must clarification within relevant timeframes in the range of 3–36 h. All these results suggest the potential industrial application of the pectinases noncovalently immobilized upon PA6 microparticles.

## 1. Introduction

Since ancient times down to the present, wine has been prepared via biochemical transformation of grape juice, known as controlled alcoholic fermentation. Most of the enzymes necessary for the fermentation to occur originate from the grapes themselves, the grapes’ own microflora and the microorganisms present during the process [1,2]. Due to the ever-growing increase in the production levels of modern winemaking, these endogenous enzymes of grapes, yeasts and other microorganisms that are present naturally in musts and wines are often neither efficient enough, nor sufficient to effectively catalyze the corresponding reactions. For this reason, the use of commercial enzymes as supplements at different stages of the winemaking process is a well-established practice in the large-scale production of wine. Nowadays, the big production volumes combined with the very demanding quality control of the final product have transformed modern winemaking into a high-tech industry that makes use of the latest achievements in biology and enzymology [3,4].

Wine production consists of four main stages [2]: (i) pressing and maceration with the purpose of extracting as much grape juice as possible for must formation; (ii) alcoholic fermentation, forming the alcohol content and many other specific features of wine; (iii) clarification, i.e., reducing the turbidity so as to achieve a clear, visually appealing and stable wine; (iv) aging and stabilization operations to optimize the physicochemical properties of the final product. All these stages play crucial roles in winemaking and their relative importance can vary depending on the style of wine being produced.

Exogenous pectinases are always needed for the clarification step as they are not produced by the yeasts used in fermentation [2]. Pectinases (E.C.3.2.1.15) are a heterogeneous enzyme group used for controlled degradation of the high-molecular-weight pectic substances contained in grape cell walls that transit into must. The main purpose of the clarification step is to reduce the must viscosity and the content of insoluble fractions in it, as well as to intensify the grape juice extraction process and maximize its yield. There are indications that pectinases can also increase the production and retention of certain volatile substances, thus releasing in wine more of the color- and flavor-determining compounds contained in grape skin [5], and making more effective the liberation of phenolic compounds [6].

Pectin is a complex substance representing methylated esters of polygalacturonic acid. Therefore, based on their mode of action, pectinases are classified as (i) deesterifying enzymes (pectin esterase) and (ii) depolymerizing enzymes (hydrolases, lyases). Depolymerizing enzymes are further classified based on their specific action site into endo- and exo-polygalacturonases and rhamnogalacturonases [7], or by the nature of the action mechanism, primary substrate and products obtained [8]. Commercial pectinase preparations should contain enzymes from each of these groups.

Apart from winemaking, pectinases are applied on a large scale in other industrial processes such as fruit juice clarification and viscosity reduction, in tomato pulp extraction, chocolate and tea fermentation, vegetable waste treatment and fiber degumming in textile and paper industries, among others. Due to their universal applications in various branches of the food industry, pectinases have a share of 25% in the global sales of food enzymes [9].

At the present time, industrial enological pectinases are most frequently used as free enzymes, i.e., soluble in the reaction medium. In this form, however, they are often unable to meet the necessary short-term operational stability and are very hard to recover and reuse [10]. Moreover, enzymatic processes with free enzymes could be difficult to make continuous and automatize [11]. To overcome these limitations, immobilization of the enzymatic catalysts can be performed. The exhaustive review by Ottone et al. [2] and other studies [12] discuss innumerous reports on diverse immobilized enzymes used at various stages of wine production. In general, immobilization of enzymes can be made on different supports (carriers) including natural or synthetic polymers, as well as inorganic substances. Physical methods (adsorption, entrapment) or chemical reactions (covalent binding to insoluble carriers, creation of crosslinked enzyme aggregates) can be implemented during immobilization. This leads to increased enzyme stability, to a final product free of biocatalysts and, finally, to the possibility of multiple uses of one and the same enzyme preparation.

In general, supports for the immobilization of enzymes that can be applied in the food industry must be nontoxic, biocompatible, insoluble under reaction conditions and mechanically and thermally stable [13]. For the particular case of pectinases’ immobilization, a group of natural biopolymers capable of easy gelation and known as alginates seem to be quite useful as supports. For example, de Oliveira et al. [14] demonstrated that enological pectinase extracted from *Aspergillus aculeatus* and immobilized in calcium alginate beads via entrapment can result in a multiple-use and thermally stable enzyme preparation applicable for clarification of fruit juices. A commercial enological pectinase (Extrazyme^®^ by Épernay, France) entrapped in alginate hydrogels [10] showed that the clarification biocatalysts so produced kept its original activity after 8 months of storage. The activity of the entrapped pectinase was retained after six reaction cycles, with 37% residual activity. Grape must turbidity decreased rapidly in the presence of this immobilized pectinase, which was more effective than the free enzyme. Immobilization of the Extrazyme^®^ enological preparation via entrapment in alginate hydrogel reinforced mechanically by addition of agar-agar was reported as a possible way to effectively and mechanically create a more resistant biocatalyst for wine clarification [15]. The pectinolytic activity of the entrapped enzyme was appropriate and it was retained after six reaction cycles, with 61% residual activity.

The search for mechanically more stable and cheaper supports readily available in various forms (granules, powders, threads, etc.) led to immobilization on synthetic polymers. Special attention was given to polyamide 6 (PA6), which is probably the closest synthetic analogue of protein biomolecules and can be obtained via ring-opening polymerization. Thus, Omelková et al. [16] and Rexová-Benková et al. [17] disclosed PA6 powders as supports of endo-polygalacturonase immobilized via covalent binding by means of glutaraldehyde. The enzyme activity decreased from 1.35 µkat/mg to 0.67 µkat/mg after immobilization, whereby the sample that was not activated by glutaraldehyde did not show any enzyme activity [16]. The changes in the action pattern of the covalently immobilized endo-polygalacturonase were ascribed to steric hindrance resulting from new covalent bonds in the enzyme molecule in the proximity of its active site [17]. More recently, Shukla et al. [18] covalently immobilized polygalacturonase from *Aspergillus Niger* on PA6 beads using a similar glutaraldehyde activation. This immobilization yielded a protein loading of 70 μg/g of PA6. The immobilized enzyme showed maximum activity at 50 °C and pH 5.0 and could be reused through four cycles of apple juice clarification with almost 50% retention of its original activity. Ben-Othman and Rinken [19] studied the immobilization of pectinolytic enzyme preparations on polyamide 66 (PA66) granules or threads applying covalent immobilization by means of dimethyl sulfate or glutaraldehyde. The immobilization yield on threads was over 40 times higher than that on pellets. The activity of the immobilized pectinase preparation for clarification of apple juice was similar to that of the free enzyme in the temperature range of 10–30 °C. The immobilized pectinase exhibited good reusability, retaining 40% of its initial activity after five successive cycles and more than 20% after twenty successive cycles. PA6 and PA66 pellets were also used by other authors as supports for pectolytic enzymes [20,21]. Covalent binding was applied in both cases through activation by dimethyl sulfate or glutaraldehyde.

There exist a number of studies on the immobilization of pectolytic enzymes on inorganic supports. Among them, the magnetic sensitive substances permitting complete and fast removal of the biocatalyst via magnetic separation possess good potential for industrial application. For example, Fang et al. [22] prepared Fe_3_O_4_ magnetic nanoparticles coated with amorphous SiO_2_ that were functionalized with NH_2_ groups. They were used as a support for covalent immobilization of pectinase with glutaraldehyde as a coupling agent. The resulting biocatalyst retained 64% of its starting activity after seven consecutive uses, and only about 21% of the total activity was lost after a 30-day storage.

Summarizing the above, the advantages of the immobilized pectinolytic enzymes are clear; however, every type of support has its own advantages and disadvantages. The polyamide supports studied so far are mechanically more robust than alginates, being also biocompatible and readily available, but they seem to require covalent immobilization. The latter, as a rule, decreases the enzyme activity and needs activation by toxic compounds, which is not acceptable in the food industry. These two problems may be resolved via effective noncovalent immobilization of the pectinases on polyamide microparticles, creating conditions for multiple hydrogen bond formations between the protein-based enzymes and a highly porous, scaffoldlike support. As shown in our previous works, such porous PA6 microparticulate supports that could also have magnetic susceptibility can be produced via activated anionic ring-opening polymerization (AAROP) of lactams [23]. The PA6 microparticles possess a suitable morphology, and their effectiveness as carriers for noncovalent immobilization of a single enzyme [24] or of enzyme dyads [25] has been demonstrated and the biocatalytic properties of these systems have been studied.

The main objective of this study was to perform noncovalent immobilization of a commercial pectinolytic preparation on PA6 microparticles with or without magnetic properties obtained via AAROP. The resulting pectinase complexes (Pec@PA6) were subjected to comparative activity and kinetic studies with the free enzyme preparation and applied for clarification of industrial rosé must. Their reusability was also investigated.

## 2. Materials and Methods

### 2.1. Materials

The ε-caprolactam (ECL) monomer with reduced moisture content for anionic polymerization (AP-Nylon^®^) was purchased from Brüggemann Chemical (Heilbronn, Germany). Before use, it was kept under vacuum for 1 h at 23 °C. As a polymerization activator, Brüggolen C20^®^ (C20) from the same company was employed. The initiator sodium dicaprolactamato-bis-(2-methoxy-ethoxo)-aluminate (Dilactamate^®^, DL, 85% solution in toluene) was purchased from Katchem (Prague, Czech Republic) and applied without further treatment. The soft, non-insulated iron particles (Fe content > 99.8%), with average diameters of 3–5 µm, were kindly donated by the manufacturer BASF, Ludwigshafen, Germany. The enological pectinase preparation Viazym Clarif Extrem^®^ derived from *Aspergillus Niger* and used for industrial clarification of white and rosé musts was a product of Martin Vialatte (Magenta, France). This pectinase preparation and sample amounts of industrial rosé musts were kindly donated by Sogrape Vinhos SA (Avintes, Portugal). The pectin from citrus peel and all other simple chemical reagents and solvents employed in this study were of analytical grade and were supplied by Sigma-Aldrich-Merck (Lisbon, Portugal).

### 2.2. Characterization Methods

Scanning electron microscopy (SEM) studies were performed with a NanoSEM-200 apparatus from FEI Nova (Hillsboro, OR, USA) using mixed secondary electron/backscattered electron in-lens detection. All the samples were observed after sputter coating with Au/Pd alloy with a 208 HR instrument from Cressington Scientific Instruments (Watford, UK) with high-resolution thickness control. The UV-VIS spectral measurements were carried out with a Shimadzu model 1900i double-beam spectrophotometer (Tokyo, Japan) working in photometric, spectral or kinetic modes. The turbidity measurements were made with a METRIA M10 portable turbidimeter (Labbox Labware, Barcelona, Spain) according to ISO 7027 performing up to 5 parallel measurements with every sample.

### 2.3. Synthesis of PA6 Microparticulate Supports

The PA6 microparticles (PA6 MPs) were synthesized via a proper method based on AAROP as previously described by Dencheva et al. [23]. First, 0.3 mol of ECL was added to 100 mL of a mixed hydrocarbon solvent (toluene/xylene 1:1 by volume) while being stirred under a nitrogen atmosphere, and the reaction mixture was refluxed for 10–15 min. Subsequently, 3.0 mol% of DL and 1.5 mol% of C20 were introduced simultaneously. The reaction time was 1 h from the point of the catalytic system addition, and the temperature was maintained in the 125–135 °C range with constant stirring at about 800 rpm. The final MPs were formed as a fine powder that was separated from the reaction mixture via hot vacuum filtration, washed several times with methanol and dried for 30 min in a vacuum oven at 60 °C. To remove the low-molecular-weight PA6 fractions, further Soxhlet extraction for 4 h with methanol was employed. The resulting neat PA6 MPs were kept in a desiccator for further treatment. The scheme of PA6 MP synthesis via AAROP is presented in Appendix A. Two types of microparticle supports were synthesized, namely neat PA6 MPs and PA6-Fe MPs. In the latter case, 3 wt.% of Fe particles (in relation to the ECL monomer) was added to the ECL/C20/DL/solvent reaction mixture, all other polymerization, isolation and purification procedures being the same as in neat PA6 MPs.

### 2.4. Pectinase Immobilization via Physical Adsorption

The Viazym manufacturer recommends a 10-fold dilution of the preparation for industrial application. For this reason, PA6 MP or PA6-Fe MP supports (100 mg) were immobilized using 5 mL of a 10-fold diluted Viazym pectinase. This solution that shows a clear absorption maximum at 263 nm (see Appendix A) was used for the subsequent protein quantifications. It was assumed that the initial 10-fold diluted Vyazim contains 10 arbitrary units (a.u.) of protein in 1 mL. The PA6 and PA6-Fe MPs were incubated at two different temperatures (23 °C or 4 °C) under gentle agitation for 24 h. Thereafter, the supernatant was decanted and stored for further analyses and use. The final Pec@PA6 and Pec@PA6-Fe complexes were washed two times with double-distilled water to remove the non-immobilized enzymes and were stored at 4 °C for further analyses.

### 2.5. Determination of the Total Amount of Protein

After isolation of Pec@PA6 and Pec@PA6-Fe at the end of the physical immobilization, the respective supernatants were subjected to UV analysis to determine the residual protein in them and calculate on this basis the amount of the total protein (TP) incorporated into the pectinase-containing complexes, expressed as
(1)$$TP=C_{0}-C_{S}\;$$
where $C_{0}$ is the starting protein content before immobilization, and $C_{S}$ is the protein content in the resultant supernatant after immobilization. The latter was determined via direct quantification of the UV absorption peak at λ ≈ 263 nm.

### 2.6. Pectinase Activity Assay

Pectinolytic activity was assayed by measuring the amount of reducing sugars released from a pectin solution in water using the DNS reagent according to the method described by Miller [26]. The DNS reagent is composed of 2-hydroxy-3,5-dinitrobenzoic acid (1% *w/v*), potassium sodium tartrate (30% *w/v*) and sodium hydroxide (1% *w/v*). In a typical assay, 0.05 mL of the commercial pectinolytic preparation (10-fold dilution) or 0.010 g of the wet pectinase-immobilized complexes were mixed with 0.45 mL substrate (0.25% *w/v* citric pectin in 50 mM citrate buffer, pH 3.8) and incubated at 50 °C for 15 min. Subsequently, 0.50 mL of DNS solution was added to the reaction mixture, and the vials with the samples were kept in water at 90 °C for 10 min and then cooled on ice. The UV absorbance of the dark yellow solution was measured at 540 nm against the neat DNS reagent [10]. The amount of released reducing sugar was quantified from a standard calibration plot constructed with D-galacturonic acid in the 0.14–2.23 µmol/mL range. One unit of pectinase activity is defined as the amount of enzyme required to release 1 μmol of galacturonic acid per minute under the assay conditions.

### 2.7. Kinetic Studies

The kinetic experiments were performed with the free commercial Viazym preparation (10-fold diluted) and its two immobilized complexes Pec@PA6 and Pec@PA6-Fe by varying the initial pectin concentration in the 0.05–5 mg/mL range and measuring the initial rates (i.e., enzyme activity) as indicated above. The total amount of enzyme added to the reaction mixture was maintained constant. Certain deviations from the conventional Michaelis–Menten kinetics were observed with the free and immobilized enzymatic complexes, i.e., substrate inhibition was present. Based on previous studies, three kinetic models, i.e., of Michaelis–Menten and two reflecting the substrate inhibition, were selected for calculation of the kinetic parameters. The functions of these models are presented in Equations (2)–(4).
(2)$$V=\frac{V_{max\;\;}S}{K_{m}+S}\;(\mathrm{Model}1)$$
(3)$$V=\frac{V_{max\;}S}{{(K}_{m}+S)(1+\frac{S}{K_{i}})}\;(\mathrm{Model}2)$$
(4)$$V=V_{max}[exp(-S/K_{i})-exp(-S/K_{m})]\;(\mathrm{Model}3)$$

The kinetic parameters (maximal rate *V_max_*, Michaelis–Menten constant *K_m_* and *K_i_*, which is a constant characterizing the formation of an inactive enzyme–substrate complex) were estimated via nonlinear multiple regression analysis using the commercial package OriginPro, version 9.8.0.200 by OriginLab corporation (Northampton, MA, USA).

### 2.8. Application of Immobilized Pectinase for Raw Grape Must Clarification

The industrially recommended concentration of free pectinase preparation or the equivalent concentrations of the Pec@PA6 or Pec@PA6-Fe complexes were mixed with 14 mL of rosé must. The turbidity of the must samples was determined in nephelometric turbidity units (NTUs) every 15 min for 3–5 h. After clarification completion, the presence of residual pectin was tested by adding acidified alcohol (5% *v/v* HCl, 5 mL) to 2.5 mL of the must sample under gentle agitation. After 10 min, the must was visually inspected. In the case of pectin-free samples, clear liquid was observed. The formation of insoluble flakes indicated the presence of residual pectin.

To evaluate the color changes in the must as a result of the enzymatic clarification, complete UV/VIS spectra of all treated must samples before and after the clarification were obtained, with their absorbances compared at 420, 520 and 620 nm.

### 2.9. Reusability Studies

The reusability of the Pec@PA6 or Pec@PA6-Fe complexes was tested by performing three consecutive clarification cycles with industrial rosé must at 23 °C, measuring turbidity every 15 min until reaching levels around 20 NTUs, according to Section 2.8. After the first clarification cycle, the two complexes were recovered from the must via centrifugation and washed two times with 1 mL of deionized water, and their clarification capacity was tested with fresh rosé must, applying the same procedure.

## 3. Results

### 3.1. SEM/EDX Characterization

As seen from the SEM micrographs in Figure 1a–c, the preliminarily synthesized empty PA6 MPs represent highly porous, scaffoldlike aggregates with sizes in the range of 100–200 μm. Each one of these aggregates comprises many fused, nearly spherical morphologies with diameters close to 20 μm. As seen at higher magnification (Figure 1c), the pores observed are open, with irregular form and different visible depths, most of them with sizes varying in the range between 200 and 800 nm. The magnetic sensitive PA6-Fe support MPs (Figure 1g–i) display very similar shapes and porosity of the microparticles. Some of the Fe-containing MPs have the form of elongated seed capsules containing several axially aligned (apparently due to magnetization) Fe particles that were coated with PA6 during the AAROP.

After immobilization of the pectinase preparation on the PA6 MPs (Figure 1d–f) and PA6-Fe MPs (1j-1l), the shape and size of the aggregates in both supports were generally preserved, as observed at lower and medium magnifications (left-hand and central columns in Figure 1, respectively). The right-hand column of images with higher magnifications, however, indicates that upon immobilization, the microporosity of both the Pec@PA6 and Pec@PA6-Fe complexes nearly disappeared. This can be attributed to filling of the pores with the enzyme after its adsorption immobilization.

More information about the composition of the pectinase-immobilized MP systems was obtained by combining SEM with energy-dispersive X-ray spectroscopy (EDX). All sites in Figure 1 where elemental analysis was performed are indicated with Z1–Z3. The respective emission spectra and the elements’ percentages are presented in Appendix A of the Appendix A. Thus, the PA6 MP support before pectinase immobilization in site Z1 (Figure 1b) displayed only the presence of C, N and O atoms, the content of N being close to 9 wt.% (Appendix A). After pectinase immobilization, elemental mapping performed in sites Z1–Z3 (Figure 1e) displayed nitrogen percentages between 17 and 24 wt.%, which was accompanied by the appearance of signals characteristic of potassium and chlorine (Appendix A). This is a direct proof of the adsorption deposition of the pectinases’ apoenzymes in the said locations, along with ions of the respective buffer solutions.

The EDX of the neat PA6-Fe MPs in site Z1 (Figure 1h) displayed the same emission spectra as the PA6 MPs. Mapping the chemical composition in the crack between the two fused PA6 spheres (site Z2 in Figure 1h) presented a weak but observable Kα line of Fe (Appendix A). After pectinase immobilization, the EDX emission in sites Z1–Z3 (Figure 1k) displayed increased amounts of nitrogen in Z2 and Z3 (see Appendix A) and the presence of K and Cl ions in all of these sites. Based on these observations, successful deposition of the pectinase can also be considered proved in the Pec@PA-Fe complex.

### 3.2. Quantification of the Enzymes in the MP-Supported Complexes

The amount of the noncovalently immobilized pectinolytic enzymes on both the PA6 MP supports was determined according to Equation (1). In Figure 2a, the UV/VIS spectra are compared for the initial Viazyme stock solution (curve 1 with the higher intensity peak) and for the remaining supernatant after removal of the immobilized pectinase-containing complexes (curves 2 and 3). This UV absorption peak centered at λ = 263 nm is attributable to the protein moieties containing a substituted benzene ring found in the pectinases’ apoenzymes, most probably originating from tryptophan, phenylalanine or tyrosine repeat units. Notably, this direct spectroscopic method produces reliable data about the adsorption-immobilized protein with a standard deviation of 3–5%, while the indirect Bradford and bicinchoninic acid colorimetric assays for protein content result in data dispersions of up to 15%.

Figure 2b displays the immobilization efficiency in the case of the Pec@PA6 and Pec@PA6-Fe complexes, which is the amount of pectinolytic enzymes noncovalently immobilized on the MP supports, as a percentage of the initial concentration in the solution before immobilization. To obtain the standard deviation values for each sample, three consecutive trials were made at every point.

As the immobilization efficiency was relatively low, i.e., about 15–20%, it was decided to investigate whether it is possible to use the same pectinase solution repeatedly, for a new immobilization. The data from Figure 2b show that this is possible, with efficiencies in the range of 15% for each consecutive immobilization cycle. The same figure also shows that immobilization is temperature-dependent. At room temperature, the immobilization process was about 5% more efficient, whereby this temperature dependence was much more pronounced in the second and each subsequent immobilization cycle.

### 3.3. Activity via the DNS Methods

It is well known that, at temperatures close to 100 °C, a reducing sugar, e.g., galacturonic acid (GA), which contains a free aldehyde group, provides a hydrogen atom to the 3,5-DNS molecule, transforming it into 3-amino-5-nitrosalycilic acid (3,5-ANS). At the same time, GA is oxidized to galactaric acid. Whenever this redox reaction occurs, a color change from yellow to dark orange is observed. This chemical interaction, on which the estimation of the pectinase activity is based, is presented in Figure 3a.

In our particular case, the free or immobilized pectinolytic preparation, due to its depolymerization action upon the pectic substrates, is expected to produce certain amounts of GA. The latter originates from the hydrolysis of the main chain of pectin, which comprises four pectic polysaccharides: homogalacturonan, xylogalacturonan and rhamnogalacturonans of type I and type II [27]. The amount of GA is strictly proportional to the 3,5-ANS concentration. The latter can be quantified via UV/VIS spectroscopy by the intensity of the band at λ = 540 nm. Clearly, then, the activity of the pectinolytic free or immobilized preparation is characterized by the concentration of 3,5-ANS released per unit time.

Figure 3b displays the evolution of the absorption at 540 nm after incubation of pectin solutions with free or immobilized Viazym preparations and subsequent treatment via the DNS method. The inset to Figure 3b represents the standard calibration curve constructed with various concentrations of GA. It is important to note here that the activity assay should be made on the basis of the absorbance at 540 nm, no matter that the maximum of the UV peak appears to be around 515–512 nm. As seen from Appendix A, the nominal 3,5-ANS peak is clearly resolved exactly at 540 nm but only at high concentrations, i.e., at higher contents of the reducing GA.

The results of the activity tests for the free and immobilized pectinolytic preparations obtained after the first and the second immobilization rounds at room temperature are presented in Figure 4. 

The Pec@PA6 and Pec@PA6-Fe samples, obtained after the first and second immobilization from the same Viazym solution, are presented together for comparison. As depicted in Figure 4a, the activities of both the Pec@PA6 and Pec@PA6-Fe complexes, from either the first or second immobilization round, closely resemble that of the free pectinolytic preparation, with the difference falling within the margin of experimental error. This means that the same pectinase starting solution can be reused many times for immobilization on fresh microparticulate carriers without deterioration in the complex enzymatic activity.

Upon normalizing these activity values with respect to the protein content (Figure 4b), it becomes evident that the specific activities of the two MP-immobilized complexes are up to 2.5 times higher than that of the free preparation. This effect contradicts the reduced activity observed in the covalently immobilized pectinase in [17], highlighting one of the advantages of noncovalent immobilization.

The notable increase in activity observed in the preparation adsorption-immobilized on PA6 MP Viazym compared with the free one could be attributed to a more advantageous active site configuration and/or the absence of inactivation in the former case.

To shed more light on this issue, comparative kinetic studies were performed with all biocatalysts of this study.

### 3.4. Kinetic Studies

The enzyme kinetics of the free pectinase preparation (Pec) and its two immobilized complexes Pec@PA6 and Pec@PA6-Fe were analyzed by plotting the dependence of the specific activity per unit protein (*V_spec_*) against the pectin substrate concentration (*S*). Figure 5a shows that for all the three catalytic systems these curves do not follow the Michaelis–Menten kinetics that would produce a saturation of *V_spec_* above a certain substrate concentration. Instead, a clear inhibition effect (i.e., diminution of *V_spec_* above 1 mg mL^−1^ of pectin) was present, best expressed with the free pectinase.

To the best of our knowledge, the above substrate inhibition in the pectinase-catalyzed pectin degradation was first established by Gummadi and Panda [28]. The authors fitted their kinetic data to five different models of substrate inhibition and established that two of them achieve best fits reaching confidence levels of 95% (Equation (3) of this study) and >99% (Equation (4)), respectively.

Equation (3) was derived by Haldane [29] and is a particular case of the more general equation of Yano et al. [30] considering the formation of multiple inactive enzyme–substrate complexes of the type ES, ES_2_…ES_n_. Equation (3) contemplates complexes with two substrate molecules per enzyme molecule and is also valid for the SES type of inactive complexes, i.e., those with allosteric inhibition with two active sites [28,31]. On the other hand, the exponential Equation (4) reflects the empirical model proposed by Edwards [32] that combines the mechanisms of diffusion-controlled substrate supply with inhibitory substrate concentration.

To ensure that fitting the experimental curves of Figure 5a to Equations (3) and (4) return *K_m_*, *V_max_* and *K_i_* values with physical meaning and good statistics, the procedure described by Cleland for one inhibitory substrate was implemented [33]. First, the kinetic data were plotted in double-reciprocal form, i.e., 1/*V_spec_* vs. 1/*S*. Then, based on the linear part of the graphs, the apparent *K_m_* and *V_max_* values were determined as in the case of Michalis–Menten kinetics (Equation (2) of this study). The missing *K_i_* constant was roughly calculated using the following relationship [33]:(5)$$K_{i}=S_{max}^{2}/K_{m}$$
wherein *S_max_* is the value of *S* at the maximum values of *V_spec_*. After obtaining these indicative values of *V_max_, K_m_* and *K_i_*, these were fed into the nonlinear fitting subroutine of OriginPro containing Equations (3) and (4), and the necessary fitting cycles were performed for each catalytic system until reaching the best adjusted R^2^ coefficient.

The best fits to the kinetic curves for free Pec, Pec@PA6 and Pec@PA6-Fe are presented in Figure 5b–d and the respective numeric data for *V_max_, K_m_* and *K_i_* are shown in Table 1.

The free Pec kinetic data were best fitted with the double exponential empiric Equation (4) that corresponds to Model 3, whereas the substrate inhibition expressed with Equation (3) and Model 2 produced an inferior fitting coefficient. This is in full agreement with the results of [28] obtained for a similar system. Thus, for the immobilized Pec@PA6 and Pec@PA6-Fe systems, only Model 3 was applied.

The *V_max_* values in Table 1 are the highest for the Pec@PA6-Fe, meaning that this complex is the fastest catalyst for pectin depolymerization. Since the *V_max_* for the Pec@PA6 is also higher than the free Pec, it can be concluded that the immobilization of Pec produces faster catalytic complexes and that the Fe particles in Pec@PA6-Fe somehow increase this effect even more. In terms of the apparent *K_m_*, the three catalytic systems display quite similar values, those of the free Pec and Pec@PA6-Fe being the largest and statistically equal, followed by that of Pec@PA6. This means that the latter immobilized complex has the highest affinity to the substrate, since it requires lower concentrations to achieve half of the *V_max_*. Finally, in terms of the *K_i_* values, the Pec@PA6 and Pec@PA6-Fe systems display similar values, more than three times higher than the free Pec. This means that the two immobilized pectinases need much higher substrate concentrations to get inhibited, as compared with the free Pec.

Summarizing the above, the noncovalent immobilization of pectinase on PA MPs results in faster catalysts, with the same or slightly higher affinity to the pectin substrate, being at the same time more difficult to inhibit.

### 3.5. Grape Must Clarification

The next step in this study was to use the Pec@PA6 and Pec@PA6-Fe catalysts for clarification of real rosé must samples containing pectin concentrations in the 2.0–2.5 mg mL^−1^ range, i.e., larger than 1 mg mL^−1^, above which the inhibition effect was registered in the kinetic studies in Figure 5. To ensure comparability of the results, the amounts of the pectinolytic preparation immobilized on the PA6 or PA6-Fe supports were kept identical to those of the free enzyme, as expressed in arbitrary units, by appropriate selection of the mass of the polyamide–pectinase conjugate. The results of all clarification studies are presented in Figure 6.

Figure 6a depicts the clarification effect at 23 °C of Pec@PA6 and Pec@PA6-Fe samples as compared with the free enzyme preparation. The two immobilized complexes show a distinct clarification effect 30 min after their introduction into the must, and after less than 120 min a full clarification with turbidity values below 20 NTUs was achieved. The free enzyme followed a similar profile, but the same clarification effect was reached 15 min later. The results in Figure 6a are in full agreement with the kinetic data (Table 1), suggesting that the Pec@PA6 and Pec@PA6-Fe should be slightly faster catalysts than the free enzyme.

Figure 6b displays the clarification capacity of Pec@PA6 and Pec@PA6-Fe conjugates after their first, second and third consecutive applications. During the first use of the complexes (curves 1 (red points) and 2 (black squares)), visually clear musts with turbidity values below 20 NTUs were obtained after approximately 3 h with both immobilized complexes. The second use of the same complexes (curve 3, blue triangles, and 4 (green triangles)) required about 16 h to reach the same turbidity, and during the third application (curves 5 and 6) the necessary clarification time was about 36 h. The reduced clarification efficiency upon repeated use can be ascribed to the possible adsorption of byproducts produced during pectin hydrolysis onto the surface of the catalytic complex. This adsorption impedes the free transport of the substrate to the active center of the enzyme, consequently elongating the time required to achieve the desired must transparency. Nevertheless, clarification durations of 16 h or even 36 h can still be considered acceptable under industrial conditions.

As regards the catalytic activity of the recycled Pec@PA6 and Pe@PA6-Fe complexes, it was studied using a standard DNS activity test during five consecutive cycles. Between the first and third cycles, the activity dropped from 100% to 60% and remained almost constant until the fifth cycle. These data are presented in Appendix A.

At the end of the clarification and reusability studies, a residual pectin test in acidified alcohol was performed for all samples, confirming that turbidity values below 20 NTUs correspond to pectin-free samples; i.e., the clarification process was finalized.

### 3.6. Color Retention of Must

Along with their transparency, the color of the rosé and white wines is also one of the most important characteristics subject to rigorous control. It is typically analyzed based on three bands in the UV/VIS spectrum: 420 nm (yellow color), 520 nm (red color) and 620 nm (blue color) [34,35]. Our comparative results of rosé musts treated with the free enzyme and the two immobilized conjugates are presented in Figure 7.

Theoretically, the use of immobilized enzymes could imply some undesirable color changes due to possible selective adsorption of wine’s color components upon the microparticulate PA6 supports, which is impossible to occur with free commercial enzyme preparation. From the UV/VIS curves in Figure 7a and from the bar graphs in Figure 7b, it can be seen that the effect of the immobilized conjugates on the absorbances at 420 nm, 520 and 620 nm is the same as that of the commercial free pectinolytic preparation. Therefore, the microparticulate polyamide supports of the Pec@PA6 and Pec@PA6-Fe conjugates do not cause any undesirable changes in the wine color, although it might have adsorbed some products of the pectin degradation, as suggested by the multiple uses of both conjugates. 

The results from Figure 7 concerning the Pec@PA6-Fe also exclude any migration (i.e., leaching) of Fe or Fe ions from the immobilized complex into the must that could lead to its specific coloration. No leaching of Fe in protein-immobilized polyamide MPs was confirmed also in an earlier work [36].

## 4. Conclusions

This is the first study on noncovalent immobilization of commercial pectinolytic preparations (Viazym) upon specially synthesized microparticulate porous PA6-based supports via AAROP.

The pectinase immobilization suggested in this work is a very simple one-pot process occurring through incubation of the microparticulate supports in buffered aqueous enzyme solutions. Morphological studies with SEM-EDX indicate that the enzyme is deposited upon the surface of the PA6 or PA6-Fe microparticles, filling their pores and channels. Each round of immobilization typically removes above 15% of the enzyme contained in the initial stock solution. This process can be performed various times with fresh support particles without a detrimental effect upon the enzymatic activity of the resulting Pec@PA complexes.

The activity studies performed with the Pec@PA6 and Pec@PA6-Fe conjugates showed that they possess a specific activity (i.e., normalized by the protein content) that is up to 2.5 times higher than that of the free enzyme. This effect was explained by comparative kinetic studies that rendered the kinetic parameters of the free enzyme and the two immobilized conjugates. It was found that the immobilization of pectinase on polyamide microparticles results in faster biocatalysts, with a slightly higher affinity to the pectin substrate, being more difficult to inhibit as compared with the respective free enzyme preparation.

The rosé must clarification studies performed at room temperature with the free pectinase and the two immobilized conjugates based on nephelometric measurements showed that complete clarification reaching 20 NTUs for all catalytic systems was achieved within 120–150 min. The reusability of the immobilized pectinase conjugates was tested in rosé must during three consecutive clarification runs. While in the first cycle complete clarification was achieved within 3 h, in the second and third cycles 16 and 36 h were needed, respectively. It should be noted that the implementation of microparticulate polyamide supports for pectinase does not change the color of the must, as measured with UV/VIS in three characteristic wavelength areas.

Currently, an enological evaluation of the wines obtained from musts clarified by means of PA6-supported enological preparations is in progress. Following this evaluation, more extensive tests will be carried out under industrial conditions using the Pec@PA6 and Pec@PA6-Fe conjugates from this study, so as to arrive at a conclusive decision regarding their potential application in the winemaking industry.

## Figures and Tables

**Figure 1 foods-13-00420-f001:**
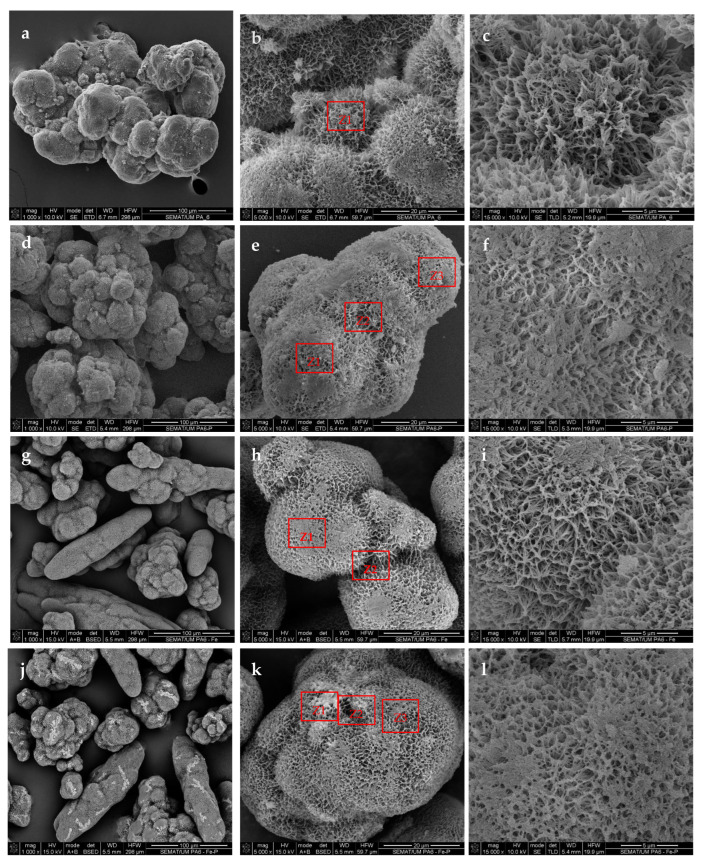
SEM micrographs of (**a**–**c**) empty PA6 MPs; (**g**–**i**) empty PA6-Fe MPs; (**d**–**f**) Pec@PA6 complexes; (**j**–**l**) Pec@PA6-Fe complexes. The EDX data for the areas designated with *Z_i_* are shown in Appendix A.

**Figure 2 foods-13-00420-f002:**
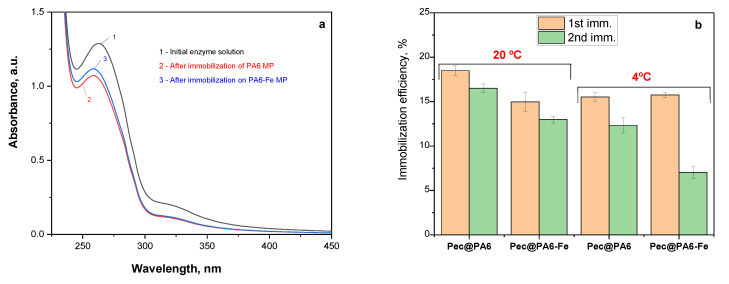
Quantification of the pectinase immobilization: (**a**) UV/VIS spectra of the Viazym solutions before and after the first round of adsorption immobilization; (**b**) immobilization efficiency for two immobilization rounds at two different temperatures for Pec@PA6 and Pec@PA6-Fe pectinolytic complexes. For more data, see the text.

**Figure 3 foods-13-00420-f003:**
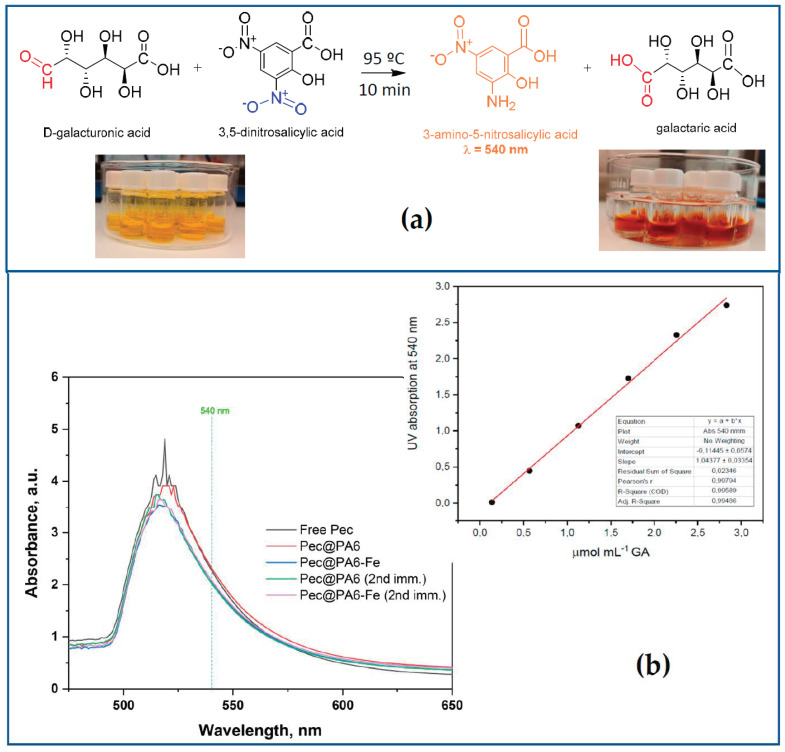
Activity assay of pectinolytic biocatalysts via the DNS method. (**a**) Reaction scheme of the interaction between 3,5-DNS reagent and D-galacturonic acid (GA); (**b**) selected full UV-VIS spectra after incubation of pectin solutions with free or immobilized pectinolytic preparations and 3,5-DNS treatment. The inset shows the standard calibration curve constructed with various concentrations of GA.

**Figure 4 foods-13-00420-f004:**
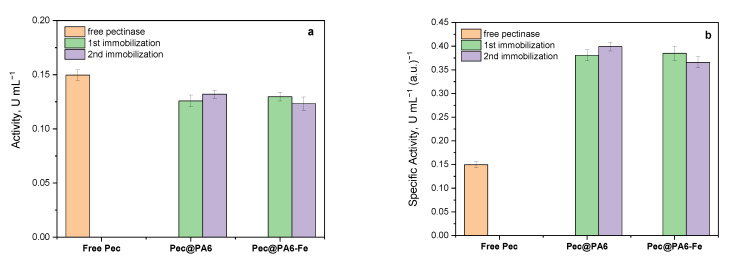
Comparative activity studies of free pectinolytic preparation and its MP-immobilized complexes: (**a**) activity in units per milliliter (U/mL) wherein 1 U is defined as the amount of enzyme required to release 1 μmol of galacturonic acid per minute under the assay conditions; (**b**) specific activity representing the activity in U.mL^−1^ per content of pectinase in arbitrary units, a.u. The standard deviation values are obtained on the basis of three trials for each of the three samples.

**Figure 5 foods-13-00420-f005:**
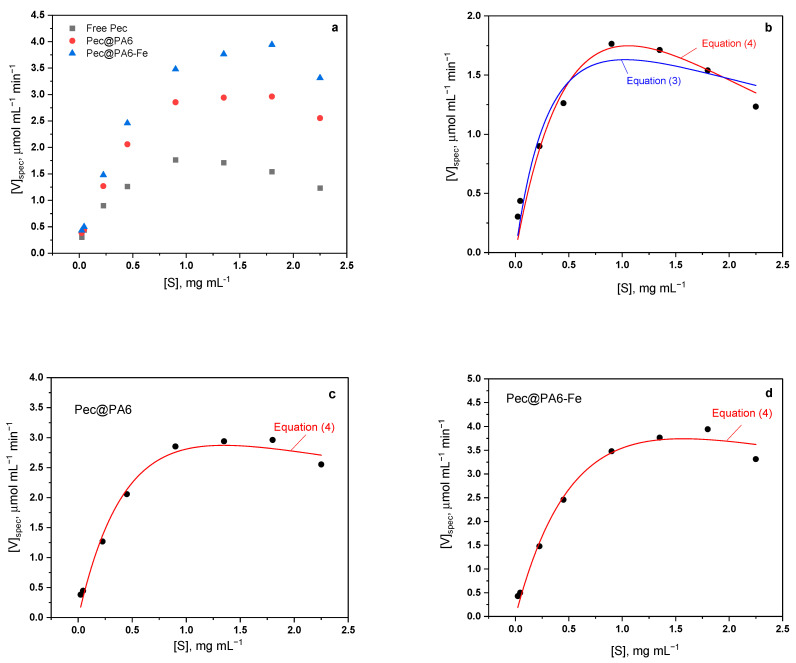
Kinetic studies with free and immobilized pectinases: (**a**) general comparison of the three kinetic curves; (**b**) free pectinase experimental curve and its best fits with Equations (3) and (4); (**c**) Pec@PA6 complex experimental curve and its best fit with Equation (4); (**d**) Pec@PA6-Fe complex experimental curve and its best fit with Equation (4). For more details, see the text.

**Figure 6 foods-13-00420-f006:**
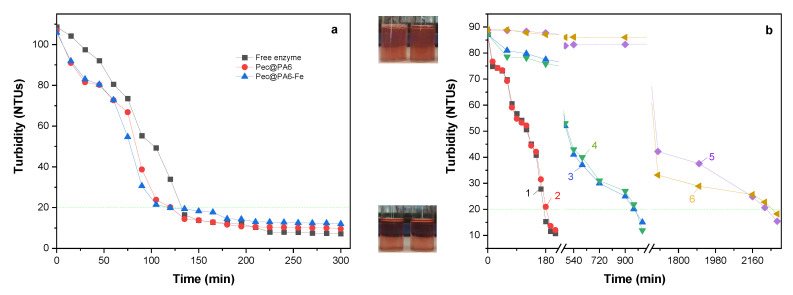
Clarification studies with free and immobilized pectinases: (**a**) clarification of must containing 2.5 mg.mL^−1^ pectinase, at 23 °C; (**b**) clarification of must containing 2.0 mg.mL^−1^ pectinase, at 23 °C; 1—Pec@PA6, first application; 2—Pec@PA6-Fe, first application; 3—Pec@PA6, second application; 4—Pec@PA6-Fe, second application; 5—Pec@PA6, third application; 6—Pec@PA6-Fe, third application. All immobilized catalysts contain 0.30 a.u. of pectinolytic enzyme, which is the amount of free enzyme employed in Figure 6a. The dashed line indicates the level of 20 NTUs below which the must is considered completely clarified. The visual aspects of the must before and after clarification are presented in the upper and lower photographs, respectively.

**Figure 7 foods-13-00420-f007:**
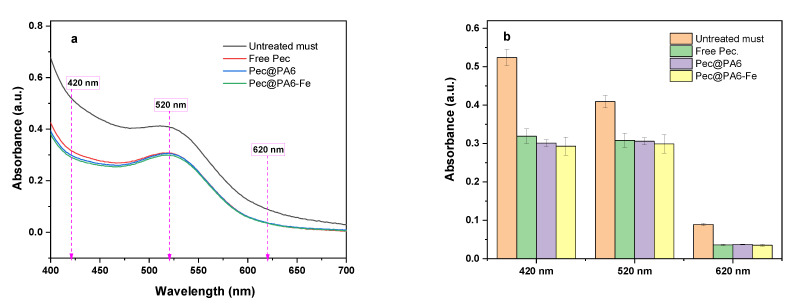
Retention of must color after clarification: (**a**) UV-VIS absorption spectra comparison in three characteristic intervals; (**b**) numerical data comparison. In the last case, the error bars are calculated on the basis of three trials for every sample.

**Table 1 foods-13-00420-t001:** Kinetic parameters estimated for Pec, Pec@PA6 and Pec@PA6-Fe.

Free Pec	Model 1	Model 2	Model 3
*V_max_* [µmol mL^−1^ min^−1^ a.u.^−1^]	2.264	4.396 ± 0.180	3.164 ± 0.113
*K_m_* [mg mL^−1^]	0.342	0.659 ± 0.233	0.518 ± 0.034
*K_i_* [mg mL^−1^]	-	1.600 ± 0.189	2.669 ± 0.163
*Adj. R^2^*	0.985 *	0.930	0.977
**Pec@PA6**	**Model 1**	**Model 2**	**Model 3**
*V_max_* [µmol mL^−1^ min^−1^ a.u.^−1^]	3.335	-	3.520 ± 0.077
*K_m_* [mg mL^−1^]	0.294	-	0.419 ± 0.028
*K_i_* [mg mL^−1^]	-	-	8.840 ± 1.111
*Adj. R^2^*	0.995 *	-	0.985
**Pec@PA6-Fe**	**Model 1**	**Model 2**	**Model 3**
*V_max_* [µmol mL^−1^ min^−1^ a.u.^−1^]	4.256	-	4.682 ± 0.108
*K_m_* [mg mL^−1^]	0.341	-	0.513 ± 0.033
*K_i_* [mg mL^−1^]	-	-	8.975 ± 1.113
*Adj. R^2^*	0.996 *	-	0.985

* *V_max_* and *K_m_* values determined from the linear part of the double-reciprocal plot.

## Data Availability

All the data generated during this research are presented in the manuscript and in the Appendix A.

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
