# Peer review of "Immobilization of Enological Pectinase on Magnetic Sensitive Polyamide Microparticles for Wine Clarification"

_foods, 2024, doi:10.3390/foods13030420_

Round 1

Reviewer 1 Report

Comments and Suggestions for Authors

Dear authors

The MS entitled “Immobilization of Enological Pectinase on Magnetic Sensitive Polyamide Microparticles for Wine Clarification.” was reviewed. The MS is well composed and the study is systematically conducted. The findings are ok. Here are few other points that should be addressed before further process.

·       Table 1 should be corrected

·       Figure 4 is not clear.

·       Why PA6 was preferred as compared to other PAs?

·        Did the authors perform any FTIR spectral measurements?

·       Does the removal rate of 15 % is correlated with some other standard?

·       What was the recycled ratio of catalysts? And how many cycles were obtained?

Comments on the Quality of English Language

English is OK

Author Response

On behalf of all coauthors, I would like to express our gratitude to the respected Reviewer 1 for the valuable opinions and constructive queries, which have significantly contributed to enhancing the quality of the manuscript. Below, we provide detailed point-by-point responses to each of the reviewers' comments.

Q1. Table 1 should be corrected.

R1: We have incorporated the Vmax, Km, and Ki dimensions into Table 1, in line with the general reviewer's suggestion to correct this table. We trust this addresses the concern.

Q2. Figure 4 is not clear.

R2: After a thorough examination, we concluded that the lack of clarity in Figure 4 may have arisen from an insufficient description. In fact, Figure 4 shows that the same pectinase stock solution can be used several consultive times (2 times shown in the figure) with fresh microparticulate immobilization carriers, without deterioration of the enzyme activity We have revised the manuscript to provide additional clarification in lines 447, 463, and 469-472.

Q3. Why PA6 was preferred as compared to other PAs?

R3: PA6 microparticles (MP) used in this study are not commercially available; we developed and patented own synthesis method for various polyamide MPs. This method is based on anionic polymerization of various lactams in solution. As a result, we have prepared PA6, PA4, PA12 MPs, as well as copolymeric MPs. In this study, the PA6 MPs were selected for their suitable morphology, (i.e., size and porosity), and the relevant amide group density along the PA6 macromolecules, ensuring effective H-bond formation between the carrier and the immobilized enzyme. Further explanation has been added in lines 145-147 and line 182 of the revised manuscript.

Q4. Did the authors perform any FTIR spectral measurements?

R4: Yes, FTIR experiments were conducted on both PA6 MP and pectinase-immobilized MPs. However, due to overlapping characteristic bands of polyamides and the enzyme proteic matrix, FTIR could not provide additional structural information.

Q5. Does the removal rate of 15 % correlate with some other standard?

R5: The term "removal rate" most probably refers to the immobilization yield of 15%. This percentage indicates that, under the specific conditions used in this submission, 15% of the enzyme from the stock solution was immobilized upon PA6 MP in the first immobilization round. Since with the same PA6 MPs but with other enzymes we have reached 80-90% immobilization yield during the first immobilization round, in the present work we reused the same stock solution with fresh PA6 MP various times. In any immobilization round, the immobilization yield was in the range of 15%. In the paper we mention results from the 1st and 2nd immobilization rounds (Figure 4). Practically, the initial stock solution can be reused for immobilization until the full exhaustion of the pectinase, i.e., up to 5-6 times.

Q6. What was the recycled ratio of catalysts? And how many cycles were obtained?

R6: We relate the above query to the reusability of our pectinase complexes. These studies were presented in Figure 6 of the initial manuscript. It shows that the pectinase-immobilized PA6 MP systems could be applied for clarification of rosé must at least in three consecutive cycles. However, the catalytic activity of these complexes can be studied also by the activity assay procedures explained in the Experimental Part, section 2.6. This activity was studied in 5 consecutive cycles. The activity during the first cycle was considered 100%, dropping to about 65% in the 3rd cycle and keeping this value until the 5th cycle. In the main manuscript text, we decided not to show these data, presenting instead the reusability in rosé must clarification. However, for fulness of the interpretation and as a consequence of the Reviewer 1 query, this study was added to the Supporting Information under the designation “Figure S5 – Relative activity of Pec@PA6 and Pec@PA6-Fe during 5 consecutive cycles”. A short paragraph presenting the essence of this discussion was added in lines 591-594 of the revised manuscript.

Reviewer 2 Report

Comments and Suggestions for Authors

This manuscript is about the use of immobilized pectinase in wine clarification, and I have some suggestions:

- The immobilization procedure used different molecular solvents (toluene, xylene and methanol). How authors guarantee that solvents residues did not go to the wine in the clarification step?

- In relation to iron particles, the could not interact with phenolic compounds, turning the wine with weird colors as green or brown?

Author Response

On behalf of all coauthors, I would like to express our gratitude to the respected Reviewer 2 for the valuable opinions and constructive queries, which have significantly contributed to enhancing the quality of the manuscript. Below, we provide detailed point-by-point responses to each of the reviewers' comments.

Q1. The immobilization procedure used different molecular solvents. How do authors guarantee that solvent residues did not go into the wine during clarification?

R1: The mentioned organic solvents were used in the synthesis of PA6 MPs, not during the immobilization stage. A comprehensive purification process, including hot filtration, multiple washing, and extended methanol extraction in a Soxhlet apparatus, ensures the removal of any solvent residues from the final MPs before immobilization. Please refer to lines 188-193 of the revised manuscript.

Q2. Could iron particles interact with phenolic compounds, altering the wine color?

R2: No evidence of iron migration or leaching into the must was observed, as confirmed by UV/VIS studies in Figure 7. It can be seen that the free pectinase and the immobilized pectinase complexes cause the same color changes at 420, 520 and 620 nm. Moreover, no leaching of Fe ions was demonstrated in previous research on protein-immobilized polyamide MPs (e.g., Dencheva et al Polymer 145:402-415 (2018)). The present manuscript has been updated with this information in lines 608-611.